# Bioengineered Skin for Diabetic Foot Ulcers: A Scoping Review

**DOI:** 10.3390/jcm13051221

**Published:** 2024-02-21

**Authors:** Nathaniel R. Primous, Peter T. Elvin, Kathleen V. Carter, Hagner L. Andrade, Javier La Fontaine, Naohiro Shibuya, Claudia C. Biguetti

**Affiliations:** 1Department of Podiatric Medicine, Surgery and Biomechanics, School of Podiatric Medicine, University of Texas Rio Grande Valley, Harlingen, TX 78550, USA; nathaniel.primous01@utrgv.edu (N.R.P.); peter.elvin01@utrgv.edu (P.T.E.); hagner.andrade@utrgv.edu (H.L.A.); javier.lafontaine@utrgv.edu (J.L.F.); naohiro.shibuya@utrgv.edu (N.S.); 2Department of Biomedical Engineering, University of Texas at Dallas, Dallas, TX 75080, USA; 3Library, School of Medicine, University of Texas Rio Grande Valley, Harlingen, TX 78550, USA; kathleen.carter@utrgv.edu

**Keywords:** tissue engineering, diabetic foot ulcers, bioengineered tissues and organs, chronic wound, tissue regeneration, bioengineered skin

## Abstract

Diabetic foot ulcers (DFUs) pose a significant threat to individuals with diabetes mellitus (DM), such as lower limb amputation and severe morbidity. Bioengineered skin substitutes (BSS) are alternatives to traditional interventions for treating DFUs, but their efficacy compared to standard wound care (SWC) or other treatment types, such as allografts, remains unknown. A scoping review of human studies was conducted to identify current approaches in the treatment of DFUs using BSS as compared with other treatment options. Systematic searches in PubMed, Cochrane Library, and Web of Science were conducted to identify comparative studies that enrolled 10 or more patients and evaluated wound healing outcomes (closure, time-to-healing, and area reduction). Database searches isolated articles published from 1 December 2012 to 1 December 2022 and were conducted in accordance with PRISMA-ScR guidelines. The literature search yielded 1312 articles, 24 of which were included for the qualitative analysis. Findings in these studies demonstrated that BSS outperformed SWC in all measured outcomes, suggesting that BSS may be a superior treatment for DFUs. Of the 24 articles, 8 articles compared human amniotic membrane allografts (hAMA) to BSS. Conflicting evidence was observed when comparing BSS and hAMA treatments, highlighting the need for future research.

## 1. Introduction

Worldwide, 425 million people were affected by Diabetes Mellitus (DM) type 1 and 2 as of 2017. DM affects approximately 10% (34.2 million adults) of the United States population [1]. Diabetic foot ulcers (DFUs) can lead to challenges in maintaining lower extremities health, and they affect an estimated 15–25% of patients with diabetes mellitus (DM) [2]. When conventional treatments for chronic wounds prove ineffective, the likelihood of soft tissue infection followed by bone infection significantly increases. Infection may lead to lower limb amputation and mortality [3]. Approximately 85% of below-the-knee amputations in DM patients are a result of non-healing DFUs [4] with a 5-year mortality of 40–50% post-amputation [2]. Furthermore, it is important to acknowledge that DFU complications not only impact morbidity and mortality but also pose a significant risk to the patient’s overall quality of life. Among those who have experienced amputation, the inevitable loss of mobility and independence will often contribute to anxiety and depression amongst that population [5]. Furthermore, non-healing DFUs and subsequent amputations not only adversely impact the patient’s physical and emotional well-being but also impose a significant financial burden. The direct financial burden of lower diabetic limb complications is conservatively estimated to exceed that of the five most costly cancers in the US, which include breast cancer at $16.5 billion and colorectal cancer at $14.14 billion annually [5].

Given the considerable challenges DFUs pose and their consequences in patients’ lives, it is imperative to prioritize research initiatives and implement strategic interventions to ameliorate DFU healing. The development of biomaterials for treating DFUs may enhance patient outcomes by mitigating amputation rates and recurrence, finally elevating the overall quality of life among diabetic individuals. Herein, we conducted a scoping review of high-quality studies to identify current approaches and potential challenges associated with the treatment of DFU using bioengineered strategies, such as Bioengineered Skin Substitutes (BSS), as compared with other options, such as extracellular matrices and standard wound care (SWC). We also present an overview of the current understanding of DFU pathophysiology, the various types of treatments available, and the biomaterials currently being utilized in this context.

## 2. Background

### 2.1. Overview of the Current Understanding of DFU Pathophysiology

DFUs are considered any below-the-ankle full thickness (deep into the dermis) chronic wound, and normally occur under advancing distal sensory neuropathy or peripheral vascular disease (PVD) [6], as demonstrated in [Figure 1]. In In brief, diabetic impairment of sensory, motor, and autonomic fibers will eventually cause the inability to perceive the protective sensations of pressure, pain, or heat and limit one’s range of motion (ROM) and mobility [4]. Together, these factors elevate plantar foot pressure and result in the formation of a “callus” or biomechanical hyperkeratotic lesion with something far more concerning underneath [6,7].

Furthermore, PVD is characterized by microvascular damage and endothelial dysfunction. Arterial basement membrane thickening and a decrease in capillary size, which are observed in diabetes, will restrict cellular exchange. Additionally, due to endothelial cell dysfunction (such as a nitric oxide synthetase deficiency), arteries and arterioles may not optimally dilate, leading to compromised blood flow and suboptimal healing [6,7].

The classification of wounds as acute or chronic is largely based on their persistence without showing signs of healing. Wounds that heal within 4 weeks are considered acute, whereas when a wound remains unhealed for more than 12 weeks, it is typically classified as chronic [8]. Of note, the cellular and molecular characteristics in acute healing versus chronic ulcers exhibit significant differences, and they are notably influenced by the inflammatory dysregulation induced by chronic hyperglycemia. 

In brief, acute foot lesions in non-diabetic individuals follow a well-orchestrated healing process consisting of four distinct yet interrelated stages: hemostasis, inflammation, proliferation, and remodeling [9]. Immediately upon injury, hemostasis begins with the release of several growth factors, such as insulin-like growth factor (IGF), epidermal growth factor (EGF), transforming growth factor-β (TGF-β), and platelet-derived growth factor (PDGF), as well as numerous cytokines from platelets after they aggregate at the site of injury and begin the conversion of fibrinogen, forming a blood clot [7,9,10]. Inflammation, the second stage, is characterized by the release of pro-inflammatory cytokines, such as tumor necrosis factor α (TNF-α) and interleukin-1 (IL-1), and the sequential recruitment of immune cells, such as neutrophils and monocytes/macrophages, to the wound site [11]. Neutrophils and macrophages are essential for removing eventual pathogens, while macrophages also clear degradation products and neutrophils [9,10]. In addition, macrophage polarization into either M1 (pro-inflammatory) or M2 (regenerative) phenotypes is a key event. M1 cells contribute to the prevention of infection and establish the immune response in the initial stages. At the same time, the M2 phenotype seems to be crucial for inflammation resolution, as well as stem cell proliferation and differentiation [12]. Lastly, the considerably overlapping stages of cell proliferation and tissue remodeling are crucial to healing completion. The proliferative phase is marked by angiogenesis, epithelialization, and the proliferation of fibroblasts for the deposition of an extracellular matrix, all mediated by regenerative factors, such as C-X-C Motif Chemokine Ligand-12 (CXCL12), TGFβ, Fibroblast Growth Factor-2 (FGF-2), and EGF [11]. This step also plays a role in epidermis regeneration by keratinocyte stem cells and the production of scar tissue, promoting wound closure [9]. Finally, M2 macrophages phagocytose excess cells during the remodeling phase and release MMPs for collagen degradation to remodel the newly formed extracellular matrix [11].

While understanding the intricacies of acute wound healing in a non-diabetic environment is crucial for managing acute foot lesions, the current knowledge of the pathophysiology of chronic wounds is key for driving diagnostic modalities and more reliable treatment alternatives. Chronic ulceration results from a deficiency or malfunction in one or more of the described healing stages [9]. First, in DFUs, there is a lack of both stem cells, which have the ability to proliferate and differentiate into multiple lineages, and growth/vascular factors, which are largely sourced from stem cells [13]. Furthermore, it has been postulated that the dysregulated synthesis of essential growth factors and cytokines in diabetic patients affects the initiation of the wound healing cascade and leads to compromised inflammatory responses and weakened host defenses [8]. These events ultimately result in delayed healing, impaired pathogenic microbe clearance, and subsequent infections [8].

Furthermore, a prolonged inflammatory stage is characterized by an imbalance in the polarization of M1/M2 macrophages, with a predominance and persistence of M1 macrophages [9]. Neutrophils accumulate and continue releasing cytotoxic granules, reactive oxygen species, and proteases, reducing the availability of crucial growth factors necessary for wound closure [14]. An overall pathophysiological aspect of the chronic DFU with non-diabetic acute wound healing is summarized in [Figure 2].

### 2.2. Types of Treatment of DFUs

The management of DFUs is a complex and challenging task for health care providers, involving multidisciplinary approaches to address various facets of the ailment, including fundamental causes of the DFU (e.g., perivascular diseases, pressure, repetitive trauma), regulation of glycemic levels, off-loading the affected area (the mainstay of DFU treatment), and mitigation of infections and other potential complications [6,15,16]. Typical treatment of a chronic foot wound begins with the preparation of the wound bed and debridement of necrotic tissue [17]. Herein, we summarize the fundamental steps for DFU wound bed preparation before proceeding to other adjuvant strategies to improve tissue healing in DFUs. For instance, the quality and preparation of the wound bed, including removing necrotic tissues or any potential infection, is essential to a successful treatment plan and outcome [18].

Debridement is a central component of wound care in DFUs [19]. It can be defined as removing necrotic or non-viable tissue and exudate, helping to reduce biofilm, minimize infection, and promote the production of granulation tissues [19,20]. The methods for debridement are diverse, falling into two main categories: selective and non-selective debridement [17].

Non-selective debridement is a conventional method that can eliminate non-viable tissue; however, some viable, healthy tissue is also often damaged and lost [17,21]. Although this may initially seem counterintuitive, the underlying concept posits that the stalled healing process can be re-initiated to traverse the protracted stage of inflammation that sustains ulceration [20,21]. This strategy is practiced in various ways, including sharp/surgical debridement, mechanical debridement via wet-to-dry dressing, ultrasonic debridement, and aqueous high-pressure lavage [17]. Sharp/surgical debridement is currently considered the gold standard, making it the reference point to which all other techniques are compared [17,22]. For this technique, instruments, such as curettes, scissors, or scalpels, are used to remove unwanted tissue and debris from the wound bed but leave what still retains the potential to heal [20,23]. Wet-to-dry debridement involves applying wet gauze moistened with saline to the wound site and allowing it to dry [17]. Desiccation causes the gauze to adhere to the wound, and when removed forcefully, it can remove both necrotic and small amounts of viable tissues [20]. The wet-to-dry method comes with inherent disadvantages of its own. Aside from this technique being unfavorably time-consuming, patients who have retained peripheral sensation report that this technique is significantly more painful than other methods of debridement. The nature of the wet-to-dry debridement technique contributes to its disadvantages of being time-consuming and reportedly quite painful for patients [20,24]. Ultrasonic debridement is another nonselective method and utilizes low-frequency ultrasonic waves of 20–40 kHz to disrupt devitalized tissue through a mechanism called cavitation, where gas bubbles rapidly expand and implode within the targeted tissue fluid [22].

Selective debridement methods include autolytic, enzymatic, and biodebridement [17]. Autolytic debridement refers to the induction of endogenous enzymes to simply digest and separate devitalized tissues from the wound bed [17]. This slow process is a component of the body’s innate ability to debride and is naturally occurring [20,23]. In contrast, enzymatic debridement is significantly faster and entails the introduction of external enzymes to necrotic tissue, facilitating their subsequent digestion [19,20,21]. Biodebridement, which is also referred to as Maggot Debridement Therapy (MDT), makes use of larvae or “maggots” from two fly species, *Lucilia sericata* and *Phaenicia sericata* [20]. The sterile larvae are placed on the wound, either directly or within a permeable biobag, and it is covered and bandaged with a permeable dressing and absorbent pad, as MDT debrided wounds tend to produce large amounts of exudate [25]. Larvae tend to lose their rate of effective activity after about 48–72 h, in which time the dressing is removed, and a washout is performed [25]. The mechanism of MDT revolves around the secretion of collagenase, serine, and chymotrypsin/trypsin-like proteases, which work together to create a nutrient-rich food source for larval consumption from non-viable tissue and biofilm while leaving healthy tissue untouched and intact [24,26]. For these reasons, MDT has shown promise as a favorable option to significantly reduce DFU bacterial load while maximizing healthy tissue preservation [24]. 

Once the wound bed is properly prepared, the healing process may return to its initial acute stage. However, it is important to note that the underlying pathology in DFUs remains, and often, achieving wound healing requires advanced therapy with adjunct materials. In the field of regenerative medicine, various strategies can be employed, including the use of skin grafts and bioengineered substitutes. 

A skin graft consists of a piece of transplanted skin positioned over an open wound with the expectation that it will adhere to and integrate into the new site. The exact mechanism by which a skin graft stimulates the growth of new skin remains unclear; however, it is believed to be accomplished by harnessing the body’s utilization of extracellular matrices, growth factors, cytokines, and stem cells inherent to the graft, all of which are vital for achieving complete wound healing [27]. With numerous options available, skin grafts can typically be categorized into four main groups: autologous grafts, allografts, xenografts, and BSS [27,28]. Autologous skin grafts must be harvested from the same individual for whom they are intended [27]. A donor site is selected based on viability, appearance, availability, and patient preference. The harvested autograft is secured over the recipient wound bed, usually by suture or dressing [29]. However, the efficacy of these grafts is susceptible to failure, often attributed to complications such as hematoma, seroma, or infection [30]. Additionally, the procedure introduces a supplementary wound at the donor site, a concern particularly relevant for patients with compromised healing abilities, such as those with DFU [31]. 

Allografts specifically involve the transplantation of tissues from a donor to a recipient. Unlike autografts, they circumvent the issue of patient donor site morbidity by sourcing the graft from a distinct human donor, commonly cadaveric or placental [32]. Because allografts originate from external sources, donors undergo screening for Human Immunodeficiency Virus, Hepatitis B and C, Cytomegalovirus, Human T-Lymphotropic Virus, Syphilis, and West Nile Virus [33]. Donor tissue is also analyzed histologically for appropriate structure, screened, and treated for immunogenicity [33,34]. Although these screening/treatment methods are extremely effective, complete microbe elimination is unlikely, and the risk of infection will persist [33]. While immuno-rejection vastly decreases after the graft undergoes pretreatment with 85% glycerol, rejection may still occur [35]. Nevertheless, allografts continue to be a viable option for chronic wound therapy.

A limitation in donor availability has led to xenografts, such as porcine skin, which involves removing sheets of animal skin, screening, and then applying directly onto a wound bed [28]. As with allografts, achieving complete microbial elimination is improbable in xenotransplantation, carrying the theoretical risk of zoonotic infection [36]. In addition to infection, hyperacute rejection, an immediate immune response, is also a clinical concern. In fact, three xeno-antigens have been identified in porcine skin alone [36]. When the body encounters xeno-antigens within the vascular endothelial cells of a graft, an antibody-mediated complement activation leads to the destruction of the transplant within hours [37].

Over the past decade, tissue engineering has advanced significantly, resulting in alternative wound care strategies such as BSS [8,38]. Theoretically, BSSs hold great potential for enhancing the healing process of DFUs by providing a scaffold for cell growth and proliferation, immunomodulation, promoting angiogenesis, and stimulating tissue regeneration [7]. BSSs aim to address the primary challenges inherent in all three preceding methods—autografts, allografts, and xenografts—namely, issues related to donor morbidity and deficiency, susceptibility to infection, and the potential for immunogenic rejection [39,40]. These types of engineered grafts can be described as skin analogs created in a laboratory, and share some of the same biological and pharmacological properties as human skin [36,41]. In this scoping review, we will initially delineate the various BSSs employed as grafting alternatives for DFUs, followed by a thorough summarization of the evidence regarding their applications in human subjects.

## 3. Materials and Methods

### 3.1. Literature Search

A comprehensive search strategy was developed to identify relevant literature on tissue engineering strategies for treating diabetic foot ulcers. Searches were conducted in accordance with PRISMA-ScR guidelines in PubMed, Web of Science, and Cochrane Library [42]. The search strategy included controlled vocabulary, Medical Subject Headings (MeSH), and common keywords for the domains of bioengineering and diabetic ulcers. The search was performed using the following keywords: “diabetic foot ulcer”, “diabetic foot”, “chronic wound”, “bioengineering”, “bioengineered skin”, “bioengineered graft”, “scaffold-based tissue engineering”, “cell-based tissue engineering”, “bioengineered skin grafts”, ”bioengineered tissue”, “skin substitute”, “dermal substitute”, “synthetic graft”, “dermal graft”, “pressure ulcer”, “foot wound”, “venous stasis ulcer”, “venostasis”, and “ischemic ulcer”. The following MeSH terms were used to improve the accuracy of the search: “Diabetic Foot/therapy” [Mesh], “Regenerative Medicine” [Mesh], “Tissue Engineering” [Mesh], “Tissue Scaffolds” [Mesh], “Skin, Artificial” [Mesh], “Skin Transplantation” [Mesh], and “Bioengineering” [Mesh]. The final search was conducted until 25 July 2023, limited to studies published in English and conducted from 1 December 2012 to 1 December 2022. Boolean operators (AND, OR, NOT) were used to combine the keywords and broaden the search. Truncation was also used to capture variations of the keywords. 

The search results were imported into Mendeley as reference management software to remove duplicates and screen for relevance. Three independent reviewers (NP, CCB, and PE) screened the titles and abstracts of all potential studies to determine eligibility for inclusion in the scoping review and to minimize bias. Subsequently, full-text articles were retrieved to verify the inclusion decision based on title and abstract screening. Any discrepancies were resolved through discussion or by a fourth reviewer (NS). Full-text articles of potentially eligible studies were obtained and assessed for inclusion using pre-defined inclusion and exclusion criteria.

### 3.2. Study Selection and Eligibility Criteria

Inclusion criteria were defined as follows: (1) clinical studies involving human subjects, (2) studies involving adult subjects, (3) clinical trials with a sample size of 10 or more patients, (4) studies evaluating bioengineered tissues for the treatment of diabetic foot ulcers, (5) studies reporting outcomes related to wound healing, such as wound closure, time-to-healing, or wound area reduction, (6) studies reporting safety and adverse events related to the use of tissue engineering strategies, (7) studies published in English, and (8) studies conducted between 1 December 2012 and 1 December 2022. Exclusion criteria consisted of (1) animal studies, (2) in vitro studies, (3) studies not related to tissue engineering strategies for treating diabetic foot ulcers, (4) studies that do not report outcomes related to wound healing, (5) studies that do not report safety and adverse events related to the use of tissue engineering strategies, (6) studies not published in English, and (7) studies conducted before 1 December 2012.

Population: a sample size of 10 or more adult diabetic patients who received bioengineered skin substitutes compared to human skin allografts or standard wound care (SWC) in treating DFUs. No exclusionary or inclusionary criteria were set for patients’ sex or the presence and severity of DFU infection.Intervention: Application of bioengineered tissues for treatment of DFUs following wound debridement.Comparison: other alternate skin substitutes or SWC methods.Outcomes: sum and percentage of wounds achieving closure, time to wound closure (median and mean), incidence of amputation (post-treatment), and adverse events related to therapy.

### 3.3. Data Extraction and Synthesis

The following data items were extracted from the included studies: year of publication, study design, cohort size, ethnicity, race, age, sex, area of ulcer, number of wounds, duration of ulcer before treatment, incidence of wound closure, percentage of wound closures, median time to wound closure, mean time to wound closure, amputation rate post-treatment, and adverse events related to therapy.

## 4. Results and Discussion

The initial literature search yielded 1312 articles: 178 from PubMed, 863 from Cochrane Library, and 271 from Web of Science [Figure 3]. After removing duplicates, 1281 abstracts were screened for their eligibility, which resulted in the exclusion of 1249 articles. The remaining 32 were placed through a full-text screening, and 8 more were removed. 24 articles were ultimately included in this review. Seven studies were potentially relevant but excluded due to the small cohort size or comparing outcomes not listed in our inclusion criteria; in [43] the authors compared a BSS to a fetal bovine collagen dressing for treating venous leg ulcers [Figure 3]. Other trials had the required number of patients; however, they did not investigate our outcomes of interest [44].

### 4.1. Types of BSS Found in this Scoping Review

The flexibility of manufacturing strategies has led to a surge of options and varieties. BSS types are often categorized by their number of layers and the content therein [Figure 4] provides a breakdown of the different BSS varieties. Some are bi-layered and acellular, such as “Integra Bilayer Wound Matrix Dressing (Integra LifeSciences Corporation, Plainsboro, NJ, USA)”, consisting of a silicone membrane as the epidermal layer and a dermis made from a combination of bovine collagen and shark chondroitin-6-sulfate glycosaminoglycan [40]. Others, like “Apligraf^®^ (Organogenesis Incorporated, Canton, MA, USA)”, are bi-layered and cellular, comprising a bioengineered dermis derived from neonatal foreskin fibroblasts and bovine type-1 collagen, along with a bioengineered epidermis from neonatal foreskin keratinocytes [40,41].

Additionally, some skin substitutes consist of just a single dermal/epidermal layer, which can be cellular or acellular. “Dermagraft^®^ (Smith and Nephew, Largo, FL, USA)”, for example, is a mono-layered cellular substitute made from a bioengineered extracellular matrix impregnated with cultured neonatal fibroblasts [41]. “Oasis^®®^ wound matrix (Smith & Nephew, Inc., Fort Worth, TX, USA)”, on the other hand, is an example of an acellular mono-layered dermal substitute [41]. It is important to note that this list is not exhaustive, as various other engineering strategies exist. Furthermore, it is crucial to understand that these skin grafts are artificially engineered rather than naturally occurring and harvested.

### 4.2. Subject Characteristics and Investigated Outcomes

All remaining studies investigated a cohort of adult patients with a history of DM and an existing DFU. Types of biomaterials and their descriptions are found in Table 1. The size and duration of pre-treatment ulceration were recorded as characteristics of the study population. Patient demographics are summarized in Table 2 and Table 3. Cohort sizes across all studies ranged from 23 to 24,823, with most patients in their fifth and sixth decades. Race or ethnicity were reported in 12 studies, and of those, seven consisted of treatment groups where over 90% of the participants were Caucasian [Table 3].Patient gender (biological sex) was documented in all studies except three, and in most cases, the majority of participants were male. 

### 4.3. Effectiveness of BSSs in Promoting Wound Closure and Improving Patient Outcomes

As mentioned above, the evaluated BSS strategies found in this search included Aligraf^®®^, Dermagraft^®®^, Oasis^®®^, Matristem Wound Matrix (Columbia, MD, USA), PriMatrix^®®^ Dermal Repair Scaffold (Integra LifeSciences Corp., Plainsboro, NJ, USA), RaphaE, Integra^®®^ Tebaderm (Teba Zist Polymer Co, Iran), AlloPatch^®®^ (Musculoskeletal Transplant Foundation, Edison, NJ, USA), PRBM (Purified Reconstituted Bilayer Matrix, Geistlich Derma-Gide, Geistlich Pharma AG), GraftJacket™ RTM (Wright Medical Group N.V., Memphis, TN, USA), DermACELL^®®^ Human Acellular Dermal Matrix (LifeNet Health, Virginia Beach, VA, USA), and Hyalograft 3D^®®^ (Fidia Advanced Biopolymers, Abano Terme, Italy). Table 1 provides the details of each BSS description and its suppliers [41,42,43,44,45,46,47,48,49,50,51,52,53,54,55,56,57,58]. The reviewed studies compared BSSs with either another BSS variant, SWC (involving sharp debridement, offloading, infection prevention, and glycemic control), or a human amniotic membrane allograft (hAMA), which was available as a dehydrated human chorion amnion membrane (dHCAM) or a viable cryopreserved placental membrane (vCPM). When BSS was compared to SWC, every recorded outcome reported BSS to be the superior treatment option with superior outcomes [41,45,58]. Comparison with hAMA yielded conflicting results. Five studies presented data favoring hAMA, indicating shorter healing times and a higher frequency of achieving complete wound closure [54,55,58,59,60] [Table 2]. Conversely, three studies reported opposing results, with BSS associated with improved healing rates and shorter healing times [43,61,62] [Table 2].

When comparing BSS varieties, there were a total of five articles, four of which compared cellular BSS to acellular BSS [41,51,63,64]. In those four, cellular BSSs demonstrated faster healing times and increased healing rates; however, they were vastly more expensive in two of those studies [51,64] [Table 2]. Cazzell et al. 2017 compared GraftJacket™ with DermACELL^®®^, two acellular dermal substitutes. In this particular trial, there were four groups, two of which received one treatment of either product (each group being assigned a BSS), while the other two groups received multiple applications of either dermal substitute (each group being assigned a BSS) [58]. Furthermore, adverse events related to treatment therapy were reported by five studies [48,49,50,55,65]. The majority of adverse events were due to infection, either osteomyelitis or cellulitis. In two studies, adverse events occurred at a higher rate in control groups that only received SWC [50,55]. Cazzell et al., 2015 reported one “maceration” of the periwound area, possibly due to the BSS treatment [49]. A summary of recorded outcomes can be found in Table 3 [Table 3].

### 4.4. Cost Effectiveness

The cost of treatment was not widely recorded among studies. In the studies comparing BSS to hAMA, only half of them assessed the cost of treatment, and all four of these studies reported that BSS resulted in significantly higher treatment costs than hAMA [55,56,59,65]. Finally, one article did not match our inclusion criteria; however, it mentioned the cost-effectiveness of these treatments and factors that can influence the extent of financial burden [66]. According to Snyder 2020, the apparent increase in treatment costs may require further evaluation since some BSSs only come in one available size regardless of the wound area. In contrast, hAMA comes in multiple sizes, which enables a smaller graft to be used as the wound contracts, thus decreasing wastage and increasing cost-effectiveness [66].

## 5. Limitations

One limitation to this scoping review is the potential exclusion of articles relevant to the topic. Studies published in a language other than English were excluded, as there are no formal translation service providers on our research team to screen these articles. In addition, while the search strategy aimed to be comprehensive, there is a possibility that studies were missed due to discrepancies in searching terminology or the databases searched. Given these limitations of the scoping review process, there may be insightful results within other studies that are not included in our synthesis of the literature. In addition, variations in the design and methodology across the included studies may introduce bias in the synthesis of the literature.

## 6. Conclusions and Perspectives

As a scoping review, the primary objective of this study was to provide an overview of the available literature on BSSs used for DFUs. As such, the focus was on the comprehensiveness of the literature rather than evaluating the quality and validity of these individual studies. We intended to identify the existing medical evidence on treatments being implemented in human subjects, providing a foundation for future research areas for further investigation. Time to wound closure and wound closure rate were the most recorded outcomes. The data suggested that any BSS might result in a superior outcome to SWC. On the other hand, BSSs, compared to hAMAs, provided contradictory results. In certain instances, the utilization of BSSs led to a reduction in wound closure time, while in some cases, BSSs yielded a lower rate of wound healing. As such, it is imperative to account for potential confounding factors in future studies.

It is also crucial to acknowledge that minor variations existed in the methods employed for SWC across different clinical trials, which might introduce a statistical bias. Notably, the choice of debridement techniques and off-loading methods, for instance, could significantly impact ulceration prognosis. Standardization of SWC across comparative studies can eliminate its potential to influence statistical outcomes.

Existing knowledge about BSSs in DFU treatment strongly suggests they have superior wound healing outcomes compared to SWC alone. Nevertheless, determination of the clinical superiority and cost-effectiveness of BSS over alternative treatment options remains inconclusive and continues to be a subject of ongoing research and comparative analysis.

## Figures and Tables

**Figure 1 jcm-13-01221-f001:**
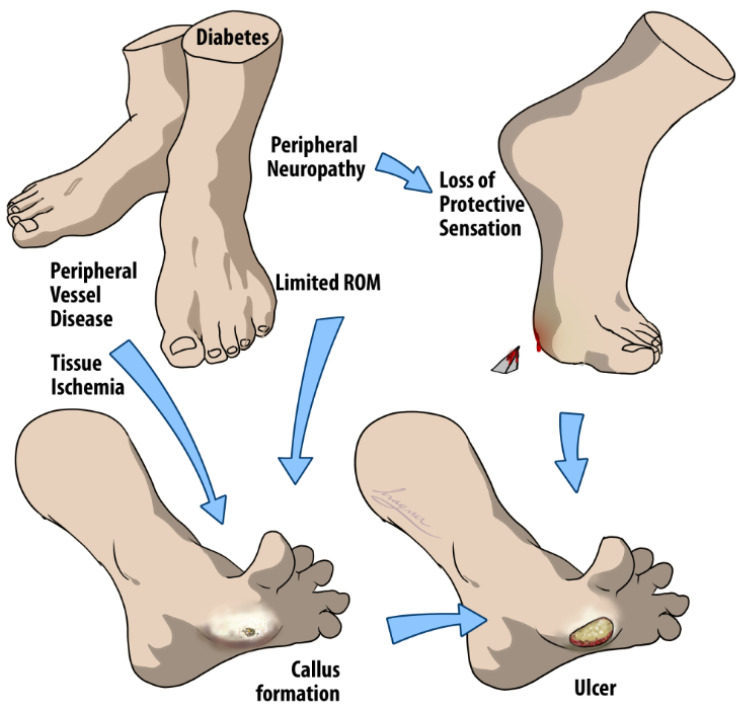
Multifactorial pathways to DFU development. The arrows in the figure suggest a progression and/or causal relationship between the conditions listed, leading to the formation of a DFU. Initially, diabetes can cause peripheral neuropathy in the feet that leads to a loss of protective sensation, preventing individuals from feeling injuries. Concurrently, diabetes may induce peripheral vessel disease, which compromises blood circulation and causes tissue ischemia, depriving tissues of necessary oxygen and nutrients. The combination of these factors, along with limited ROM, contributes to abnormal pressure distribution on the feet. This abnormal pressure often results in callus formation, which, if left unchecked, can result in open wounds due to the body’s diminished ability to heal itself effectively, thus completing the multifactorial process of DFU development.

**Figure 2 jcm-13-01221-f002:**
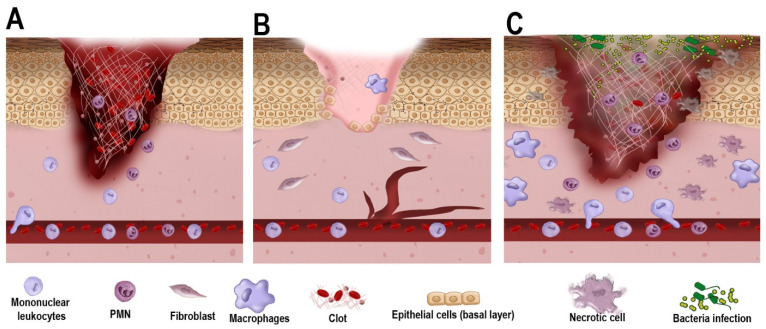
Cellular and molecular mechanisms of wound healing in acute vs. chronic wound healing. (**A**). As trauma occurs, the acute wound undergoes blood clot formation with the early infiltration of polymorphonuclear (PMN) cells and mononuclear leukocytes (e.g., monocytes). The blood clot serves as a scaffold for cell migration and the temporary closure of the wound. (**B**). In a non-diabetic environment, as healing progresses, fibroblasts promote wound contraction, and epithelial cells from the basement membrane proliferate towards wound closure. Macrophages contribute to removing debris and secrete growth factors that further aid in wound closure. (**C**). In a diabetic environment, owing to poor vascularization and dysregulated inflammatory response, cell proliferation and wound contraction are delayed. PMN and macrophages also exhibit a defective response, favoring bacterial infection, necrosis, and impaired healing.

**Figure 3 jcm-13-01221-f003:**
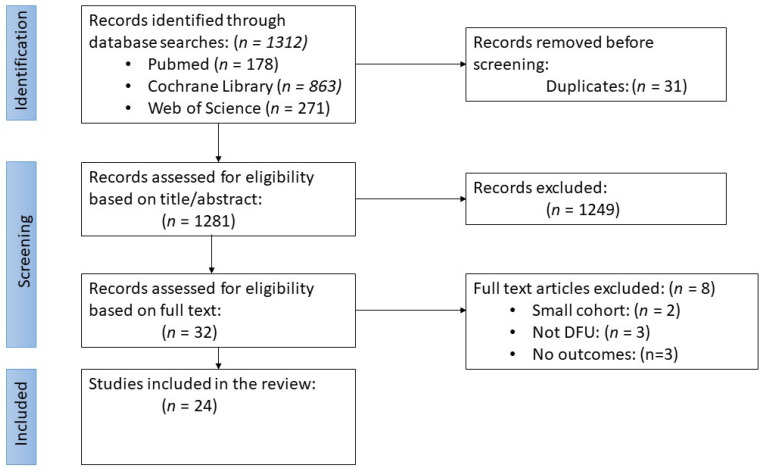
Flow diagram for the literature search and study selection process according to the PRISMA-ScR guidelines.

**Figure 4 jcm-13-01221-f004:**
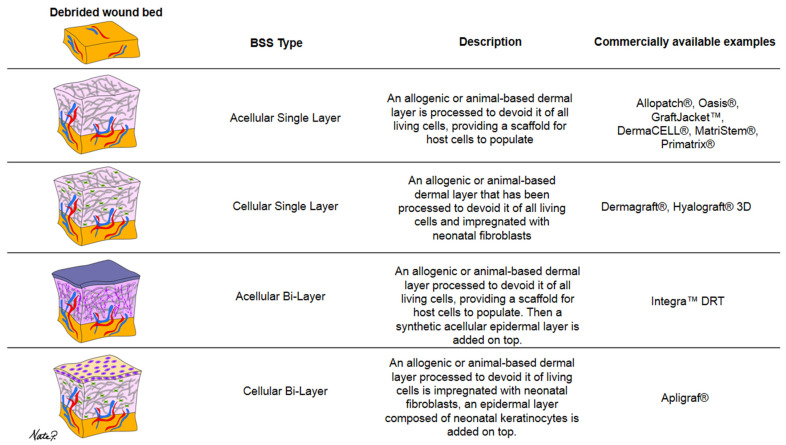
Categorization of BSS varieties by number of layers and content description.

**Table 1 jcm-13-01221-t001:** Types of BSS reported in the literature.

Study	Product Name	Company	Product Description
Armstrong et al., 2022	Derma-Gide^®^	Geistlich Pharma North America, Inc., Princeton, NJ, USA	Bi-layered acellular matrix derived from porcine material
Campitiello et al., 2017	Integra Flowable Wound Matrix^®^	Integra LifeScience, Corp., Princeton, NJ, USA	Bi-layered acellular matrix from bovine tendon collagen and glycosaminoglycan (chondroitin-6-sulfate)
Cazzell et al., 2015, Gilligan et al., 2015, Tchangque Fossuo et al., 2019	OASIS^®^ Extracellular Matrix	Cook Biotech Inc., West Lafayette, IN, USA; exclusively marketed by Smith and Nephew, Inc., Memphis, TN, USA	Tri-layer porcine small intestine submucosa, acellular, collagen-based extracellular matrix
Djavid et al., 2020	Tebaderm^®^ Collagen Matrix	Tebaderm manufacturer, Treetta Advanced Wound Care Products., Mashhad, Iran	Collagen matrix dressing including chitosan/collagen hydrogel
Driver et al., 2015	Omnigraft^®^ Dermal Regeneration Matrix	Integra LifeScience, Corp., Princeton, NJ, USA	Bi-layered with bioengineered Silicone and Collagen/Chondroitin-6-sulfate matrix
Rosa et al., 2019	RAPHA^®^ System–Latex biomembrane	Department of the Industrial Complex and Innovation in Health (DECIIS) and the Engineering and Innovation Laboratory (LEI/UnB)., Brasilia, Federal District, Brazil	Natural latex biomembrane originating from Hevea brasiliensis
You et al., 2014	Hyalograft 3D	CHA BIO&DIOSTECH CO LTD, Seongnam, South Korea	Autologous skin fibroblasts in 3D scaffolds formed of hyaluronic acid derivatives
Zelen et al., 2014, Zelen et al., 2015, Kisner et al., 2015, Kraus et al., 2017, Glat et al., 2019	Apligraf^®^	Organogenesis, Inc., Canton, MA, USA	Bi-layered cellular skin substitute composed of human neonatal fibroblasts cultured in a bovine type I collagen matrix over human neonatal epidermal keratinocytes.
Zelen et al., 2014, Zelen et al., 2015, Kisner et al., 2015, Kraus et al., 2017	EpiFix^®^	MIMEDX Group, Inc., Marietta, GA, USA	Composed by placental tissue allograft containing human amnion/chorion membrane
Zelen et al., 2016, Zelen et al., 2018	AlloPatch^®^ Pliable	MTF Biologics, Corp., Edison, NJ, USA	Human acellular dermal matrix
Ananian et al., 2018, Sabolinski et al., 2019	Grafix Prime^®^	Osiris Therapeutics, Inc., Columbia, MD, USA	Viable cryopreserved human placental membrane
Glat et al., 2019	AmnioBand^®^ Membrane	MTF Biologics, Corp., Edison, NJ, USA	Human acellular placental matrix
Sanders et al., 2014, Gilligan et al., 2015, Frykberg et al., 2015, Ananian et al., 2018, Sabolinski et al., 2019, Fitzgerald et al., 2019, Tchangque Fossuo et al., 2019	Dermagraft^®^	Organogenesis, Inc., Canton, MA, USA	Three-dimensional polyglactin mesh substrate containing human neonatal dermal fibroblasts
Sanders et al., 2014	TheraSkin^®^	LifeNet Health, Virginia Beach, VA, USA	Human extracellular matrix containing viable human fibroblasts and keratinocytes
Cazzell et al., 2017	DermACELL^®^	MATRACELL^®^ technology from LifeNet Health., Virginia Beach, VA, USA	Human acellular tissue matrix allograft
	GraftJacket^®^	LifeCell, Corp., Branchburg, NJ, USA	Human acellular dermal matrix
Fitzgerald et al., 2019	Primatrix^®^	Integra LifeScience, Corp., Princeton, NJ, USA	Fetal bovine acellular dermal matrix
Frykberg et al., 2016	MatriStem^®^	Acell, Inc., Columbia, MD, USA	Urinary Bladder Matrix

**Table 2 jcm-13-01221-t002:** Clinical Outcomes and Cost for DFUs treatment modalities.

Study	Group Distribution per Type of Treatment Modality	Number of DFUs	Wound Closure (n, %)	Time to Wound Closure	Cost Per-Patient	Amputation, n (%)	Treatment Related Adverse Events (n, %)
Armstrong et al., 2021	Group 1 Adv	12,676	NR	NR	NR	Minor: 490 (3.9%) Major: 197 (1.6%)	NR
	Group 2 Adv FPFU Group 2 Adv No FPFU	1131 1131				Minor: 22 (1.9%) Major: <11 (<1%) Minor: 51 (4.5%) Major: 18 (1.6%)	
	SWC Group 1	12,676	NR	NR	NR	Minor: 551 (4.3%) Major: 402 (3.2%)	NR
	SWC Group 2	1131				Minor: 47 (4.2%) Major: 30 (2.7%)	
Armstrong et al., 2022	Derma-Gide^®^	20	17 (85%)	Mean: 37 days	$1731	NR	NR
	SWC	20	6 (30%)	Mean: 67 days	SWC: NR	NR	NR
Campitiello et al., 2017	Integra	23	20 (86.95%)	Mean: 29.73 days	NR	10 (43.48%)	10 (43.48%)
	SWC	23	12 (53.17%)	Mean: 42.78 days	NR	15 (65.2%)	15 (65.2%)
Cazzell et al., 2015	OASIS^®^ Extracellular Matrix	41	22 (54%)	Median: 9 weeks	NR	NR	1 (2.4%)
	SWC	41	13 (32%)	Median: 11 weeks	NR	NR	0 (0%)
Djavid et al., 2020	Tebaderm^®^ Collagen Matrix	30	18 (60%)	Median: 11.8 weeks	NR	NR	2 (6.67%)
	SWC	31	11 (35.5%)	Median: 21.4 weeks	NR	NR	3 (9.68%)
Driver et al., 2015	Omnigraft^®^ Dermal Regeneration Matrix	154	79 (51%)	Median: 43 days	NR	NR	7 (4.55%)
	SWC	153	49 (32%)	Median: 78 days			8 (5.23%)
Frykberg et al., 2015	Dermagraft^®^	163	NR	NR	NR	9 (5.5%)	9 (5.5%)
	SWC	151				19 (12.6%)	19 (12.6%)
Rosa et al., 2019	RAPHA^®^ System–Latex biomembrane with HCP	12	4 (66.6%)	NR	NR	NR	NR
	RAPHA^®^ System–Latex biomembrane without HCP	8	2 (25%)				
	SWC	5	1 (20%)				
You et al., 2014	Hyalograft 3D	31	26 (84%)	Mean: 36.4 days	NR	NR	No adverse events related to the study dressings
	SWC	32	11 (34%)	Median: 48.4 days			
Zelen et al., 2014	Apligraf^®^	20	9 (45%)	Median: 49 days	Mean: $9216.00	NR	No adverse events related to the study dressings
	EpiFix^®^	20	19 (95%)	Median: 49 days	Mean: $1669.00		
	SWC	20	7 (35%)	Median: 49 days	NR		
Zelen et al., 2015	Apligraf^®^	33	24 (73%)	Mean: 47.9 days, Median:NR	Mean: $8918	NR	No adverse events related to the study dressings
	EpiFix^®^	32	31 (97%)	Mean:NR, Median: 23.6 days	Mean: $2798		
	SWC	35	18 (51%)	Mean: 57.4 days, Median: NR	NR		
Zelen et al., 2016	AlloPatch^®^ Pliable	20	13 (65%)	Mean: 40 days	Mean: $1475	NR	No adverse events related to the study dressings
	SWC	20	1 (5%)	Mean: 77 days	Median: $963		
Zelen et al., 2018	AlloPatch^®^ Pliable	40	AlloPatch: 34 (68%)	Mean: 38 days	Mean: $1200	NR	No adverse events related to the study dressings
	SWC	40	SWC: 12 (30%)	Mean: 72 days	Median: $680		
Ananian et al., 2018	Grafix Prime^®^	31	15 (48.4%)	Mean: 38 days	$3846.25	NR	3 (9.7%)
	Dermagraft^®^	31	12 (38.7%)	Mean: 31 days	$7968.75		10 (32.26%)
Glat et al., 2019	AmnioBand^®^ Membrane	30	27 (90%)	Mean: 32 days	Mean: $2900.00	NR	No adverse events related to the study dressings
	Apligraf^®^	30	12 (40%)	Mean: 63 days	Mean: $9700.00		
Kirsner et al., 2015	Apligraf^®^	163	72%	Median: 13.3 weeks	NR	NR	NR
	EpiFix^®^	63	47%	Median: 26 weeks			
Kraus et al., 2017	Apligraf^®^	59	76%	Median: 12 weeks	NR	8.9% of all patients enrolled in the study underwent amputation or bone resection	NR
	EpiFix^®^	63	50%	Median: 19.4 weeks			
Sabolinski et al., 2019	Dermagraft^®^	1444	61%	Median: 20 weeks	NR	NR	No significant differences between groups in adverse events
	Grafix Prime^®^	178	46%	Median: 36 weeks			
Sanders et al., 2014	Dermagraft^®^	12	4 (33.3%)	Mean: 12.5 weeks	NR	N/A	No adverse ulcer related events were observed
	TheraSkin^®^	11	7 (63.6%)	Mean: 8.9 weeks			
Cazzell et al., 2017	DermACELL^®^	53	29 (54.7%)	NR	NR	Subjects with amputations due to infection were excluded from the protocol	Not speficic adverse events repored
	GraftJacket^®^	23	9 (39.13%)				
	SWC	56	30 (53.57%)				
Fitzgerald et al., 2019	Dermagraft^®^	108	69 (64%)	Median: 14.6 weeks	NR	NR	NR
	Primatrix^®^	100	43 (43%)	Median: 25 weeks			
Frykberg et al., 2016	MatriStem^®^	27	7 (25.9%)	Mean: 69.8 days	$1780.63	9 (33.3%)	No adverse events were observed with procedure or product related
	Dermagraft^®^	29	9 (31.0%)	Mean: 65.7 days	$11,371.43	19 (65.5%)	
Gilligan et al., 2015	OASIS^®^ Extracellular Matrix	13	10 (77%)	Mean: 36 days	$2522	NR	NR
	Dermagraft^®^	13	11 (85%)	Mean: 41 days	$3889		
Tchangque Fossuo et al., 2019	Dermagraft^®^	17	8 (47.1%)	NR	NR	NR	None of the adverse events were related to the procedures and products
	OASIS^®^ Extracellular Matrix	19	14 (73.7%)				
	SWC	19	11 (57.9%)				

SWC = Standard of wound care, RAPHA^®^ System-Latex Membrane; HCP means applied by a Health Care Professional. For a complete list of BSS products refer to Table 1. * Armstrong et al., 2021 used Group 1 Adv (Advanced treatment comprised of cellular and acellular dermal substitutes).

**Table 3 jcm-13-01221-t003:** Demographic and Clinical Characteristics of Study Participants with DFUs.

Study	Treatment Modality	Age, Mean (SD)	Race: n (%)	Ethnicity: n (%)	Sex: n (%)	Wound Duration	Area of Ulcer (cm^2^)
Ananian et al., 2018	Grafix Prime^®^	55.13 (12.09)	White: 32 (84.2%)	Hispanic: 22 (57.9%)	Male: 28 (73.7%)	Mean: 199.32, Median: 191 days	Mean: 7.15, Median: 5.0
			Black/African American: 3 (7.9%)				
			American Indian/ Alaskan Native: 1 (2.6%)	Non-Hispanic: 16 (42.9%)	Female: 10 (6.3%)		
			Other: 2 (5.3%)				
	Dermagraft^®^	58.1 (11.89)	White: 34 (91.9%)	Hispanic: 21 (56.8%)	Male: 32 (86.5%)	Mean: 146.32, Median: 125 days	Mean: 5.7, Median: 5.0
			Black/African American: 1 (2.7%)				
			American Indian/ Alaskan Native: 0 (0%)	Non-Hispanic: 16 (43.2%)	Female: 5 (13.5%)		
			Other: 2 (5.4%)				
Armstrong et al., 2021 *	Group 1 SWC	70.8 (11.7)	White: 10,226 (81.7%)	NR	Male: 7296 (58.3%)	NR	NR
			Black: 1589 (12.7%)				
			Hispanic: 273 (2.2%)				
			Native American: 122 (1.0%)		Female: 5214 (41.7%)		
			Asian: 72 (0.6%)				
			Other: 123 (1.0%)				
			Unknown: 105 (0.8%)				
	Group 1 Adv	70.7 (11.5)	White: 10,122 (82.2%)	NR	Male: 7268 (59.0%)	NR	NR
			Black: 1342 (10.9%)				
			Hispanic: 373 (3.0%)				
			Native American: 122 (1.0%)				
			Asian: 84 (0.7%)		Female: 5045 (41.0%)		
			Other: 137 (1.1%)				
			Unknown: 133 (1.1%)				
	Group 2 SWC	71.4 (11.4)	White: 933 (82.0%)	NR	Male: 661 (58.0%)	NR	NR
			Black: 142 (13.0%)				
			Hispanic: 22 (2.0%)				
			Native American: 34 (3.0%)				
			Asian: 34 (3.0%)		Female: 470 (42.0%)		
			Other: 34 (3.0%)				
			Unkown: 34 (3.0%)				
	Group 2 Adv FPFU	71.9 (11.2)	White: 954 (84.0%)	NR	Male: 643 (58.0%)	NR	NR
			Black: 109 (10.0%)				
			Hispanic: 36 (3.0%)				
			Native American: 32 (3.0%)				
			Asian: 32 (3.0%)		Female: 488 (44.0%)		
			Other: 32 (3.0%)				
			Unkown: 32 (3.0%)				
	Group 2 Adv No FPFU	70.8 (11.6)	White: 929 (82.4%)	NR	Male: 678 (60.1%)	NR	NR
			Black: 126 (11.2%)				
			Hispanic: 39 (3.5%)				
			Native American: 34 (3.0%)		Female: 450 (49.9%)		
			Asian: 34 (3.0%)				
			Other: 34 (3.0%)				
			Unkown: 34 (3.0%)				
Armstrong et al., 2022	Derma-Gide^®^	59.3 (13.35)	Caucasian: 20 (100%)	NR	Male: 13 (65%)	Mean SD: 12.1 (8.21) weeks	Mean SD: 2.5 (2.16)
			African American: 0 (0%)		Female: 7 (35%)	Median (IQR): 9 (8) weeks	Median (IQR): 1.7 (1.4)
	SWC	66.5 (11.26)	Caucasian: 19 (95%)	NR	Male: 12 (60%)	Mean SD: 15.6 (12.96) weeks	Mean SD: 3.5 (2.85)
			African American: 1 (5%)		Female: 8 (40%)	Median (IQR): 8 (17) weeks	Median (IQR): 3.0 (3.8)
Campitello et al., 2017	Integra Flowable Wound Matrix^®^	64.04 ± 8.94	NR	NR	Male: 15 (65.22%)	Mean: 38.56 (12.61) weeks	NR
					Female: 8 (34.78%)		
	SWC	62.08 ± 7.71	NR	NR	Male: 13 (56.53%)	Mean: 39.5 (9.90) weeks	NR
					Female: 10 (43.47%)		
Cazzell et al., 2015	OASIS^®^ Extracellular Matrix	57.1 (10.9)	White: 33 (81%)	Hispanic/Latio: 10 (24%)	Male: 32 (78%)	Mean (SD): 21.3 (12.3) weeks	Mean (SD): 2.1 (2.3)
			Non-white: 8 (20%)	Non-Hispanic/Latino: 31 (76%)	Female: 9 (22%)	Median (Min–Max): 19.0 (7.0–49.0) weeks	Median (Min–Max): 1.2 (0.3–10.5)
	SWC	56.6 (10.8)	White: 33(81%)	Hispanic/Latio: 16 (39%)	Male: 30 (73%)	Mean (SD): 22.2 (13.5)	Mean (SD): 2.6 (7.5)
			Non-white: 8 (20%)	Non-Hispanic/Latino: 25 (61%)	Female: 11 (27%)	Median (Min–Max): 18.0 (7.0–49.0)	Median (Min–Max): 1.0 (0.4–48.4)
Cazzell et al., 2017	DermACELL^®^	59.1 (12.176)	NR	NR	Male: 57 (80.3%)	Mean (SD): 40.0 (71.56) weeks	Mean (SD): 3.9 (4.15)
					Female: 14 (19.7%)	Median: 20.1 (6.0–479.0) weeks	Median: 1.9 (1.0–21.0)
	SWC	56.9 (10.86)	NR	NR	Male: 51 (73.9%)	Mean (SD): 36.4 (38.84) weeks	Mean (SD): 3.6 (3.61)
					Female: 18 (26.1%)	Median: 15.3 (2.0–167.0) weeks	Median: 2.30 (1.0–20.0)
	GraftJacket^®^	58.5 (9.83)	NR	NR	Male: 20 (71.4%)	Mean (SD): 36.8 (53.60) weeks	Mean (SD): 3.3 (2.69)
					Female: 8 (28.6%)	Median: 13.5 (2.0–226.0) weeks	Median: 2.00 (1.0–11.0)
Djavid et al., 2020	SWC	57.3 (13.2)	NR	NR	Male: 22 (71%)	NR	Mean: 3.5 (4.2)
					Female: 9 (29%)		Median (range): 2.0 (0.5–22)
	Tebaderm^®^ Collagen Matrix	54.2 (13.2)	NR	NR	Male: 18 (60%)	NR	Mean: 3.09 (2.5)
					Female: 12 (40%)		Median (range): 2.5 (0.5–12)
Driver et al., 2015	Omnigraft^®^ Dermal Regeneration Matrix	55.8 ± 10.6	White: 118 (76.6%)		Male: 118 (76.6%)	Omnigraft: Days	Omnigraft: cm^2^
			African American: 28 (18.2%)	NR		Mean (SD): 308 (491)	Mean (SD): 3.53 (2.5)
			Hispanic: 46 (29.9%)			Median (IQR): 126 (288)	
	SWC	57.3 ± 9.7	White: 111 (72.5%)		Male: 114 (74.5%)	SWC: Days	SWC: cm^2^
			African American: 34 (22.2%)	NR		Mean (SD): 303 (481)	Mean (SD): 3.65 (2.7)
			Hispanic: 83 (27.0%)			Median (IQR): 152 (224)	
Fitzgerald et al., 2019	Dermagraft^®^	60.2	NR	NR	Male: 78/106 (74.3%)	Mean (SD): 8.8 (11.7) months	Mean (SD): 4 (3.6)
	Primatrix^®^	65.2	NR	NR	Male: 66/100 (66.0%)	Mean (SD): 12.8 (48.3) weeks	Mean (SD): 5.8 (4.6)
Frykberg et al., 2016	MatriStem^®^	57.0 (9.8)	Caucasian: 22 (81.5%)	Hispanic or Latino: 10 (37%)	Male: 21 (77.8%)	NR	4.3 (5.7)
			Non-Caucasian: 5 (18.5%)	Non-Hispanic or Latino: 17 (63%)	Female: 6 (22.2%)		
	Dermagraft^®^	58.5 (11.4)	Caucasian: 25 (86.2%)	Hispanic or Latino: 25 (82.2)	Male: 22 (75.9)	NR	3.2 (4.5)
			Non-Caucasian: 4 (13.8%)	Non-Hispanic or Latino: 4 (13.8)	Female: 7 (24.1)		
Gilligan et al., 2015	OASIS^®^ Extracellular Matrix	62.2 (12.2)	NR	NR	Male (%): 76.9%	Minimum of 4 weeks	1.9 (1.8)
	Dermagraft^®^	63.4 (9.8)	NR	NR	Male (%): 61.5%		1.9 (1.4)
Glat et al., 2019	AmnioBand^®^ Membrane	62 (13.2)	Caucasian: 28 (93%)	NR	Male: 16 (53%)	Mean (SD): 12.3 (14.25)	Mean (SD): 2.4 (1.88)
			African American: 2 (7%)		Female: 14 (47%)	Median: 7.5 weeks	Median: 1.4
	Apligraf^®^	62 (15.28)	Caucasian: 27 (90%)	NR	Male: 23 (77%)	Mean (SD): 14.5 (14.7)	Mean (SD): 3.1 (2.29)
			African American: 3 (10%)		Female: 7 (23%)	Median: 8 weeks	Median: 2.1
Kirsner et al., 2015	Apligraf^®^	Mean (SD): 60.1 (12.5)	NR	NR	Male: 104 (68%)	Mean (SD): 4.4 (2.6)	Mean (SD): 6.0 (5.5)
		Median: 60			Female: 49 (32%)	Median: 3.8 months	Median: 3.9
	EpiFix^®^	Mean (SD): 61.1 (12.2)	NR	NR	Male: 48 (76.2%)	Mean (SD): 4.6 (3.0)	Mean (SD): 5.2 (5.0)
		Median: 62			Female: 15 (23.8%)	Median: 3.5 months	Median: 3.0
Kraus et al., 2017	Apligraf^®^	61	NR	NR	Male: 13	Mean (SD): 4.2 (2.5)	Mean (SD): 4.8 (5.1)
					Female: 46	Median: 3.7 months	Median: 2.7
	EpiFix^®^		NR	NR	Male: 15	Mean (SD): 4.6 (3.0)	Mean (SD): 5.2 (5.0)
					Female: 48	Median: 3.5 months	Median: 3.0
Sabolinski et al., 2019	Dermagraft^®^	62 (12.4)	NR	NR	Male: 472 (32.7%)	Mean (SD): 8.97 (14.96)	Mean (SD): 7.23 (7.74)
					Female: 878 (68.3%)	Median: 4.63 months	Median: 4.0
	Grafix Prime^®^	62 (12)	NR	NR	Male: 45 (25.4%)	Mean (SD): 11.54 (19.30)	Mean (SD): 6.43 (6.73)
					Female: 131 (75.6%)	Median: 6.13 months	Median: 3.6
Tchangque Fossuo et al., 2019	Dermagraft^®^	62.83 (9.03)	Caucasian: 17 (100%)	Hispanic: 2 (11.8%)	Male: 17 (100%)	Mean (SD): 37.61 (96.07) weeks	Week1: 1.60 (1.79)
			Non-Caucasian: 0 (0%)	Non-Hispanic: 15 (88.2%)	Female: 0 (0%)		Week 12: 0.33 (0.57)
							Week 28: 0.29 (0.69)
	OASIS^®^ Extracellular Matrix	61.88 (8.64)	Caucasian: 16 (84.2%)	Hispanic: 2 (10.5%)	Male: 18 (94.7%)	Mean (SD): 10.91 (7.56) weeks	Week 1: 3.08 (3.79)
			Non-Caucasian: 3 (15.8%)	Non-Hispanic: 17 (89.5%)	Female: 1 (5.3%)		Week 12:0.31 (0.76)
							Week 28:0.08 (0.36)
	SWC	63.31 (9.09)	Caucasian: 18 (94.7%)	Hispanic: 0 (0%)	Male: 17 (89.5%)	Mean (SD): 21.68 (36.06) weeks	Week 1:1.29 (0.90)
			Non-Caucasian: 1 (5.3%)	Non-Hispanic: 19 (100%)	Female: 2 (10.5%)		Week 12:0.20 (0.38)
							Week 28:0.05 (0.10)
You et al., 2014	Hyalograft 3D	61.2 (11.4)	NR	NR	Male: 21 (68%)	Mean (SD): 6.1 (16.4)	Mean (SD): 3.5 (3.7)
					Female:10 (32%)	Median (Min–Max): 4.4 (1.5–84.0) months	Median (Min–Max): 1.7 (1.0–15.6)
	SWC	63.8 (10.7)	NR	NR	Male: 22 (69)	Mean (SD): 6.2 (19.7)	Mean (SD): 2.9(2.7)
					Female: 10 (31)	Median (Min–Max): 3.9 (1.5–108.0) months	Median (Min–Max): 2.1 (1.0–14.3)
Zelen et al., 2014	Apligraf^®^	65.2 (11.7)	Caucasian: 18 (90%)	NR	Male: 9 (45%)	Mean: 18.5 (13.8)	Mean: 2.6 (1.8)
			African American: 2 (10%)		Female: 11 (55%)	Median (min, max): 13 (6, 54) weeks	Median (min–max): 2.1 (1.0–6.8)
	EpiFix^®^	63.2 (13.0)	Caucasian: 19 (95%)	NR	Male: 10 (50%)	Mean: 15.6 (12.7)	Mean: 2.7 (2.4)
			African American: 1 (5%)		Female: 10 (50%)	Median (min, max): 11 (5, 54) weeks	Median (min, max): 2.0 (1.0, 9.0)
	SWC	62.2 (12.8)	Caucasian: 17 (85%)	NR	Male: 9 (45%)	Mean: 16.2 (13.5)	Mean: 3.3 (2.7)
			African American: 3 (15%)		Female: 10 (45%)	Median (min, max): 9 (6, 52) weeks	Median (min, max): 2.0 (1.0, 9.0)
Zelen et al., 2015	Apligraf^®^	63.8 (11.86)	Caucasian: 30 (29.7%)	NR	Male: 14 (13.9%)	Mean: 19.0 (14.78)	Mean: 2.7 (2.75)
			African American: 3 (30%)		Female: 19	Median: 16 (4,52)	Median: 1.7 (1.0–14.7)
	EpiFix^®^	63.3 (12.25)	Caucasian: 31 (30.7%)	NR	Male: 19 (18.3%)	Mean: 17.3 (15.3)	Mean: 2.6 (2.97)
			African American: 2 (2.0%)		Female: 13	Median (min, max): 12 (3, 52)	Median: 1.7 (1.0, 16.9)
	SWC	60.6 (11.55)	Caucasian: 31 (30.7%)	NR	Male: 22 (21.8%)	Mean: 14.1 (12.9)	Mean: 3.1 (3.17)
			African American: 3 (3%)		Female: 13	Median (min, max): 8 (2, 50)	Median: 1.8 (1.0, 15.5)
Zelen et al., 2016	AlloPatch^®^ Pliable	61.5 (10.85)	Caucasian: 20 (100%)	NR	Male: 16 (80%)	NR	Mean (SD):4.7 (5.24)
			African American: 0 (0%)		Female: 4 (20%)		
	SWC	57.1 (10.65)	Caucasian: 19 (95%)	NR	Male: 12 (60%)	NR	Mean (SD):2.7 (2.26)
			African American: 1 (5%)		Female: 8 (40%)		
Zelen et al., 2018	1st AlloPatch^®^ Pliable	62(11)	White: 20 (100%)	NR	Male: 16 (80%)	Mean Healing Time: 6–12 weeks	4.7 (5.3)
			African Americans: 0 (0%)		Female: 4 (20%)		
	2nd AlloPatch^®^ Pliable	55 (13)	White: 16 (80%)	NR	Male: 12 (60%)		1.7 (0.61))
			African Americans: 4 (20%)		Female: 8 (40%)		
	1st SWC	57 (11)	White: 19 (95%)	NR	Male: 12 (60%)		2.7 (2.3)
			African Americans: 1 (5%)		Female: 8 (40%)		
	2nd SWC	67 (14)	White: 19 (95)	NR	Male: 12 (60%)		2.6 (2.7)
			African Americans: 1 (5%)		Female: 8 (40%)		
Frykberg et al., 2015	Dermagraft^®^ SWC	NR	NR	NR	NR	Greater than 6 week duration	NR
Sanders et al., 2014	Dermagraft^®^	56.58 (14.96)	White Non-hispanic: 66.67% Black: 33.33%	NR	Male: 6 (50%) Female: 6 (50%)	Mean: 11.71 (8.02) weeks	4.78 (3.95)
	TheraSkin^®^	60 (15.74)	White Non-hispanic: 54.55% Black: 45.45%	NR	Male: 5 (45.45%) Female: 6 (54.55%)	Mean: 43.58 (78.08) weeks	5.45 (5.58)
Rosa et al., 2019	RAPHA^®^ System–Latex biomembrane with HCP RAPHA^®^ System–Latex without HCP	NR	NR	NR	NR	Wound Advent Period: 2 months–10 years	Wound size: 1.5–299.14 cm^2^

SWC= Standard of wound care, RAPHA^®^ System-Latex Membrane; HCP means applied by a Health Care Professional. For a complete list of BSS products refer to Table 1. * Armstrong et al., 2021 used Group 1 Adv (Advanced treatment comprised of cellular and acellular dermal substitutes), FPFU = Followed Parameters for Use.

## Data Availability

The authors will provide the raw data supporting the article’s conclusion without any unnecessary hesitation.

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
