# Peer review of "Bioengineered Skin for Diabetic Foot Ulcers: A Scoping Review"

_jcm, 2024, doi:10.3390/jcm13051221_

Round 1
Reviewer 1 Report
Comments and Suggestions for Authors
The authors prepared a comprehensive review about the clinical outcome using bioengineered skin substitutes for DFU. As review manuscript the review deals about the clinical application of bioengineered skin for DFU. The authors compare published literature regarding clinical outcome. Very relevant is the overview of used bioengineered skin and the clinical outcome done by extensive literature search and well prepared tables. The overview compares different bioengineered skin used clinical and shed light that the outcome is not always the same compared to the different centers using the bioengineered skin. Methodology used is fine. Conclusion are fine based on the search strategy/comparisons. Listed references are appropriate. Quality of tables is fine. Abbreviations in the tables might be explained, depends on the journal regulations.
Author Response
Thank you for your comments. Abbreviations were added to the table legend. However, some abbreviations, such as names and descriptions of the BSSs are presented in Table 1. In this way, we added a table legend on Tables 2 and 3 to refer to Table 1.
Reviewer 2 Report
Comments and Suggestions for Authors
1. Lines 225, 250: Please, correct typos.
2. Tables 1 and 2 were submitted as supplementary files, but they should appear in the main document. Please, revise.
3. The information reported in Tables 1 and 2 evidences a careful and meticulous data extraction process conducted by the authors, based on the 24 selected publications. Nonetheless, from a reader’s standpoint, the formatting of these tables requires significant improvement, due to the current visual crowding effect (especially for Table 2). In other words, it would be extremely helpful to the reader if the information in each main row (corresponding to an individual study) was further compartmentalized to enhance visual discrimination of the information, and thus, reading comprehension. This would in turn make the presented compilation more impactful and meaningful. In this sense, adding subdivisions to each main row would easily allow the reader to identify what information refers to what treatment (SWC or BSS). Please, revise.
4. Please, consider including Supplementary Table 1 (Types of BSS reported in the literature) in the main text of the manuscript, as it provides highly relevant background information to the reader.
Author Response
- Lines 225, 250: Please, correct typos.
Ans: Thank you for your careful observation. We have corrected the repeated sentence in the line 225, but we did not identify typos in the sentence 250.
- Tables 1 and 2 were submitted as supplementary files, but they should appear in the main document. Please, revise.
Ans: Thank you for your observation. To clarify, Tables 1 and 2 are indeed integral components of the article and were accidently submitted as a supplementary, it has been corrected. With the inclusion of the Supplementary Table 1 as part of the main article, it has been redesignated as Table 1. Consequently, the original Table 1 has been renumbered as Table 2, and what was previously Table 2 is now Table 3.
- The information reported in Tables 1 and 2 evidences a careful and meticulous data extraction process conducted by the authors, based on the 24 selected publications. Nonetheless, from a reader’s standpoint, the formatting of these tables requires significant improvement, due to the current visual crowding effect (especially for Table 2). In other words, it would be extremely helpful to the reader if the information in each main row (corresponding to an individual study) was further compartmentalized to enhance visual discrimination of the information, and thus, reading comprehension. This would in turn make the presented compilation more impactful and meaningful. In this sense, adding subdivisions to each main row would easily allow the reader to identify what information refers to what treatment (SWC or BSS). Please, revise.
Ans: We agree with the reviewer and we have improved the spacing and visual of the tables.
- Please, consider including Supplementary Table 1 (Types of BSS reported in the literature) in the main text of the manuscript, as it provides highly relevant background information to the reader.
Ans: Thank you for your suggestion, we have implemented this change.
Reviewer 3 Report
Comments and Suggestions for Authors
background
Page 2 lines 64-65: peripheral vascular disease (PVD)[6]. [Figure 1] illustrates this mechanism.
Please correct this sentence as follows: peripheral vascular disease (PVD) [6], as demonstrated in [Figure 1].
Page 3 line 73: Figure 1. Multifactorial pathways to DFU development. Please remove this sentence as it adds nothing to the literature and to avoid repetition, because [figure 1] is mentioned previously.
Page 5 line 205: The title [Current grafting alternatives for the management of DFUS]. From line 206 to line 279 including Figure 4, please summarize this dialog and write the whole details in the methodology section of the main manuscript. No more details are required in the background section of the main manuscript.
Author Response
Page 2 lines 64-65: peripheral vascular disease (PVD)[6]. [Figure 1] illustrates this mechanism.
Please correct this sentence as follows: peripheral vascular disease (PVD) [6], as demonstrated in [Figure 1].
Ans: Thank you for your careful correction. This change has been implemented.
Page 3 line 73: Figure 1. Multifactorial pathways to DFU development. Please remove this sentence as it adds nothing to the literature and to avoid repetition, because [figure 1] is mentioned previously.
Ans: Thank you for your careful review and valuable feedback. While we deeply appreciate your suggestions, we respectfully disagree with the proposed change regarding the title "Multifactorial Pathways to DFU Development." This decision was made to ensure consistency with Figure 2, which is titled "Cellular and Molecular Mechanisms of Wound Healing in Acute vs. Chronic Wounds." We believe that retaining the original title provides a clearer and more direct link to the content presented in the figure, thereby offering a more comprehensive and explanatory insight into our research findings. We hope this rationale clarifies our decision to maintain the title as is
Page 5 line 205: The title [Current grafting alternatives for the management of DFUS]. From line 206 to line 279 including Figure 4, please summarize this dialog and write the whole details in the methodology section of the main manuscript. No more details are required in the background section of the main manuscript.
Ans: Thank you for the thoughtful feedback. In response, we have diligently worked to refine our manuscript for clarity and conciseness. The introductory section now concludes at line 259 with a statement outlining our approach: " In this scoping review, we will initially delineate the various BSSs employed as grafting alternatives for DFUs, followed by a thorough summarization of the evidence regarding their applications in human subjects."
Subsequent content, originally extending from lines 260 to 280 and detailing our findings, has been integrated into the results section. This adjustment not only aligns with the reviewer's suggestion but also enhances the logical flow of our presentation, ensuring that the results are clearly delineated from the introductory overview. Furthermore, to accommodate this restructuring, we have adjusted the figure sequence, reassigning what was previously Figure 3 to now be presented as Figure 4. This change better organizes the visual aids in alignment with the revised narrative flow of the text. Our revision strategy was guided by the goal of making the review accessible and informative for readers unfamiliar with the topic, while still preserving the thoroughness and precision of our analysis. We believe these adjustments significantly improve the manuscript by making it more reader-friendly without sacrificing the integrity of the scientific discourse.